# Age-Dependent Effect of Calcitriol on Mouse Regulatory T and B Lymphocytes

**DOI:** 10.3390/nu16010049

**Published:** 2023-12-22

**Authors:** Agata Śnieżewska, Artur Anisiewicz, Katarzyna Gdesz-Birula, Joanna Wietrzyk, Beata Filip-Psurska

**Affiliations:** Department of Experimental Oncology, Hirszfeld Institute of Immunology and Experimental Therapy, Polish Academy of Sciences, 53-114 Wroclaw, Poland; agata.maria.pawlik@gmail.com (A.Ś.); a.anisiewicz@gmail.com (A.A.); kat.gdesz@gmail.com (K.G.-B.)

**Keywords:** aging, calcitriol, regulatory T lymphocytes, Tregs, regulatory B lymphocytes, Bregs, splenocytes, adipose tissue, lymph nodes, blood

## Abstract

The hormonally active vitamin D_3_ metabolite, calcitriol, functions as an important modulator of the immune system. We assumed that calcitriol exerts different effects on immune cells and cytokine production, depending on the age of the animal; therefore, we analyzed its effects on regulatory T lymphocytes and regulatory B lymphocytes in healthy young and old female C57Bl/6/Foxp3GFP mice. In the lymph nodes of young mice, calcitriol decreased the percentage of Tregs, including tTregs and pTregs, and the expression of GITR, CD103, and CD101; however, calcitriol increased the level of IL-35 in adipose tissue. In the case of aged mice, calcitriol decreased the percentages of tTregs and CD19+ cells in lymph nodes and the level of osteopontin in the plasma. Additionally, increases in the levels of IgG and the lowest levels of IFN-γ, IL-10, and IL-35 were observed in the adipose tissue of aged mice. This study showed that calcitriol treatment had different effects, mainly on Treg phenotypes and cytokine secretion, in young and old female mice; it seemed that calcitriol enhanced the immunosuppressive properties of the lymphatic organs and adipose tissue of healthy young mice but not of healthy aged mice, where the opposite effects were observed.

## 1. Introduction

The aging process is associated with numerous functional and phenotypic changes in T and B lymphocytes and monocytes/macrophages. In addition, elderly people have an increased incidence of autoimmune and infectious diseases as well as a higher risk of cancer [1]. For example, naïve CD4^+^ CD45RA^+^ cells are replaced by memory CD4^+^ CD45RA^−^ cells with increasing age. Furthermore, an increased proportion of cells co-expressing the CD57 molecule was found in the CD8^+^ T lymphocyte subgroup [2]. Additionally, the antibody-mediated immune response is qualitatively compromised by the production of antibodies exhibiting lower affinity and autoreactivity [3]. The generation and function of regulatory T lymphocytes (Tregs), which differentiate in the thymus (tTregs) or extrathymically in the peripheral sites (pTregs), are also regulated in an age-dependent manner [4]. Salminen et al. postulated that myeloid-derived suppressor cells (MDSCs), whose population significantly increases with age, serve as enhancers of other immunosuppressive cells, such as Tregs and regulatory B cells (Bregs), and as potent inducers of immunosenescence [5].

Vitamin D_3_ deficiency is common around the world and is associated with severe consequences, particularly in older people [6]. Depending on the level of 25-hydroxycholecalciferol (25(OH)D_3_) in the serum, the so-called nonclassical effects of vitamin D are counted as disorders, the frequency or severity of which increase with age; these include various metabolic disorders, increased risk of sepsis, and cardiovascular diseases [6]. The hormonally active vitamin D_3_ metabolite, calcitriol (1,25-dihydroxyvitamin D_3_), functions as an important modulator of the immune system [7]. Interestingly, almost all cells of the immune system express the vitamin D receptor (VDR) [8]. Immune cells also express 1-hydroxylase enzyme CYP27B1 and can therefore generate calcitriol [9,10].

Bregs are immunosuppressive cells that support immune tolerance. By producing interleukin (IL)-10, IL-35, and transforming growth factor (TGF)-β, Bregs inhibit immunopathological states by interfering with the expansion of pathogenic T lymphocytes and other proinflammatory lymphocytes [11]. The effects of calcitriol on B cells include the inhibition of recombination and class-switching ability, a decrease in immunoglobulin secretion, and an increase in IL-10 production. In addition, naïve B lymphocytes incubated with calcitriol ex vivo exhibited a decreased ability to co-activate and consequently reduce the expansion and production of cytokines by naive T lymphocytes [12]. Previous studies have also revealed that the secretion of IFN-γ and IL-2 by T cells decreased following exposure to calcitriol whereas the production of IL-4, IL-5, and IL-10 increased, resulting in a shift toward Th2/Tregs responses. In addition, calcitriol induces the Th2 response by increasing the expression of the GATA3 transcription factor but promotes the production of Tregs by upregulating Foxp3 transcription [13,14,15]. Another study of young mouse models showed that calcitriol inhibited the penetration of Th1 and Th17 lymphocytes into the spinal cord and increased the proportion of Tregs (CD4^+^ CD25^+^ Foxp3^+^ and CD4^+^ IL-10^+^) and Bregs (CD19^+^ CD5^+^ CD1d^+^) in the peripheral lymphatic organs, which inhibits the development of experimental autoimmune meningitis [16]. Similarly, calcitriol increased the percentage of Tregs (CD4^+^ FoxP3^+^) and Bregs (CD19^+^ IL-10^+^) and reduced proinflammatory Th17 (CD4^+^ IL-17^+^) cells in mice with immunization-induced epidermolysis bullosa acquisita [17]. On the other hand, in young mice bearing 4T1 mammary gland cancer, the calcitriol analog PRI-2191 stimulated Th17 cell differentiation and IL-17 secretion [18], thereby promoting metastasis and angiogenesis [19]. On the other hand, contrasting effects (reduced secretion of IL-17 [18] and antimetastatic effect [20]) were observed in old ovariectomized mice. In another study of patients (34–55 years old) with pituitary gland tumors, exposure to calcitriol was found to suppress the expression of IL-10 in B cells. The authors found that VDR was bound by the IL-10 transcription factor, which interfered with the expression of IL-10 in B cells [21].

In our previous studies of mice bearing 4T1 mammary gland tumors, we showed that calcitriol and its analogs may have different effects on immune cells depending on the age of the mice (6–8 weeks old vs. 60 weeks old, ovariectomized), including their phenotypes, production of cytokines, and the expression of inflammatory process associated markers—acute phase proteins. Due to these effects, we observed increased lung metastasis of breast cancer cells in young mice [19], whereas a transient suppression of metastasis was observed in old mice [20]. Moreover, an increased immune response toward Th2 and Tregs was seen in the lymph nodes (LNs) of young mice (indicated by the expression of a number of genes) [22]. Increased percentages of B lymphocytes (CD19^+^) [22] and increased levels of TGF-β in blood plasma [19] were observed in our studies of young tumor-bearing mice treated with vitamin D analogs, whereas the number of CD19^+^ cells [18] and the level of TGF-β in old mice were not affected [20]. Additionally, we observed an increased percentage of CD3^+^ CD4^+^ CD25^+^ lymphocytes in splenocytes from 4T1 tumor-bearing young mice [22], whereas calcitriol decreased these lymphocytes significantly in old mice [18]. However, 4T1 tumor progression was accompanied by a severe inflammatory response [23] that may in turn affect the response to the treatment.

Based on our research results and data available in the literature, we assume that calcitriol exerts different effects on immune system cells and cytokine production depending on the age of the animal [19,20,22]. However, most related studies were conducted in animals afflicted by various diseases that influence the immune system. No studies in the literature have focused on the differences in the immune responses of healthy young and old mice to treatment with vitamin D in the context of Tregs and Bregs. Thus, in the present study, we analyzed the effects of calcitriol on Tregs and Bregs in healthy young (6–12 weeks) and old (~1 year) female C57Bl/6/FoxP3^GFP^ mice. 

## 2. Materials and Methods

### 2.1. Mice

The study was carried out in 18 6-week-old (about 18 g) and 18 52-week-old (about 25 g) female C57Bl/6/Foxp3^GFP^ mice. The experiments were approved by the Local Ethical Committee for Experiments on Animals in Wroclaw, Poland (Approval Code: resolution No. 96/2017, Approval Date: 25 October 2017, application titled “Does the age of the body affect the induction of regulatory T and B lymphocytes under the influence of vitamin D3—preliminary studies” was submitted on 10 October 2017 to the Local Ethical Committee for animal experiments in Wrocław by IIET PAS), and were conducted in accordance with the Directive of the European Parliament and Council No. 2010/63/EU on the protection of animals used for scientific purposes (Based on Article. 48 point 1 of the Act of 15 January 2015 on the protection of animals used for scientific or educational purposes. Journal of Laws, item 266). The mice were obtained from the Animal Facility of the Ludwik Hirszfeld Institute of Immunology and Experimental Therapy, Polish Academy of Sciences, Wroclaw, Poland. They were maintained under a 12-h day/night cycle with unrestricted access to feed and drinking water in the SPF (specific pathogen-free) area in the Institute’s animal facility.

We used C57Bl/6/Foxp3^GFP^ mice expressing green fluorescent protein (GFP) under the control of the Foxp3 gene promoter [24] to evaluate the effects of calcitriol on Tregs and Bregs in healthy young and old C57Bl/6/Foxp3^GFP^ study mice. The intensity of GFP fluorescence, which was proportional to the level of Foxp3 transcription and allowed the identification of different Treg populations, was analyzed in these mice using flow cytometry [24].

### 2.2. Design of the Experiment

Each group of mice (18 young mice and 18 old mice) was divided into two groups: calcitriol (Cayman Chemicals, Ann Arbor, MI, USA) treatment group and control group (*n* = 9). The whole experiment consisted of 4 groups: two control groups (control-young, 9 mice; control-old, 9 mice) and two treatment groups (calcitriol-young, 9 mice; calcitriol-old, 9 mice). The mice were treated subcutaneously thrice a week with calcitriol at a dose of 0.5 µg/kg for 3 weeks and then euthanized. Polyethylene glycol (80%) (Sigma-Aldrich, Saint Louis, MO, USA) was used as a vehicle control. The body weights of all the mice were monitored throughout the experiment. Blood, spleens, LNs (inguinal and axillary lymph nodes), and abdominal visceral adipose tissue were collected for analysis.

### 2.3. Tissue Preparation

Whole blood samples were collected from the experimental mice and placed in Eppendorf tubes (with 80 μL heparin at 5000 I.U./mL; Polfa Tarchomin, Warsaw, Poland). The blood samples were centrifuged at 4 °C for 15 min at 2000× *g* to obtain plasma, which was subsequently stored at −80 °C and used in further investigations. To obtain mononuclear cells, blood cells were suspended in Hank’s Balanced Salt Solution (IIET PAS, Wroclaw, Poland) and centrifuged at 400× *g* for 40 min at room temperature using gradient density centrifugation (Ficoll Paque Premium 1.084; Sigma-Aldrich, Saint Louis, MO, USA). Mononuclear cells were collected, washed in PBS solution, centrifuged, and then immediately used for cytometric staining.

Spleen and LN samples were collected from young and aged mice using sterile surgical instruments and transferred to RPMI-1640 medium containing HEPES (Gibco, Scotland, UK), 2% FBS (Gibco, Scotland, UK), and antibiotics (100 U/mL penicillin and 100 μg/mL streptomycin; Sigma-Aldrich, Saint Louis, MO, USA). A single-cell suspension of splenocytes and LN cells was prepared by passing them through sterile nylon filters (70 μm) on a petri dish and then centrifuging twice at 192× *g* and 4 °C. Spleen samples were further treated with Red Blood Cell Lysis Buffer (Sigma-Aldrich, Saint Louis, MO, USA) for 1 min at a 1:1 ratio, and then centrifuged and resuspended in PBS buffer. The single-cell suspensions thus obtained were immediately used for cytometric staining.

Additionally, splenocytes (1 × 10^6^ cells/mL) were stimulated with phorbol 12-myristate 13-acetate (PMA; Sigma-Aldrich, Saint Louis, MO, USA) at 25 ng/mL for 4 h, and the supernatants were collected for further analyses.

Adipose tissue was homogenized immediately after collection in RIPA buffer containing protease and phosphatase inhibitors (1:100; all Sigma-Aldrich, Saint Louis, MO, USA). Briefly, the tissue was placed in homogenization tubes with ceramic spheres (MP Biomedicals, Irvine, CA, USA). RIPA buffer was added to the tubes, and homogenization was carried out for 2–3 cycles in a FastPrep-24 Instrument (MP Biomedicals, Irvine, CA, USA). The prepared homogenate was incubated on ice for 20 min, centrifuged twice (12,000× *g*, 15 min, 4 °C), and then frozen at −80 °C for further analysis.

### 2.4. Blood Morphological Parameter Analysis

Blood samples collected from the mice were analyzed using a Mythic 18 hematology analyzer (PZ Cormay S.A., Lomianki, Poland).

### 2.5. ELISA Tests

The plasma, supernatant, and adipose tissue samples were subjected to DC Protein assay (Bio-Rad, Hercules, CA, USA) to determine their protein concentrations.

The expressions of osteopontin (OPN), IFN-γ, TGF-β, IL-10, 17β-estradiol, 25(OH)D_3_, and IgG in the plasma, supernatants of stimulated splenocytes, and adipose tissue were analyzed using ELISA kits according to the manufacturers’ protocols. The results were read out using a Synergy H4 Hybrid reader (Bio-Tek Instruments Inc., Winooski, VT, USA). A standard curve was prepared based on the absorbance of standard solutions of known concentrations; this was then used to determine the concentrations of the test samples.

The following mouse ELISA kits were used for analyses: IgG, IL-35, and OPN (Elabscience, Houston, TX, USA); 25(OH)D_3_ and E2 (MyBiosource, San Diego, CA, USA); IFN-γ, TGF-β, and IL-10 (eBioscience, currently Thermo Fisher Scientific, Waltham, MA, USA).

### 2.6. Flow Cytometry Analyses

The single-cell suspensions prepared from spleens, LNs, and blood mononuclear cells isolated from young and aged mice (1 × 10^6^) were resuspended in PBS with 2% FBS (GE Healthcare, Chicago, IL, USA) and incubated with CD16/CD32 antibodies to block the Fc receptors. The live/dead cells were determined by staining using the Zombie UV Viability Kit (Biolegend, San Diego, CA, USA) according to the manufacturer’s instructions. After incubation, splenocytes were stained with anti-mouse conjugated antibodies for multicolor flow cytometry analysis (Table 1; isotype controls were included as well). Cells were stained separately to analyze the characteristics of T and B cells. The following Breg subpopulations were analyzed (as the literature reports that different markers are used to identify Bregs, we decided to use a wide panel of markers in our study):MZB cells: CD19^+^ CD21/CD35^+^ CD1d^+^T2-MZP cells: CD19^+^ CD21/CD35^+^ CD1d^+^ IgM^hi^ CD24^hi^ CD23^hi^ IgD^hi^B10 cells: CD19^+^ CD21/CD35^+^ CD1d^+^ IgM^hi^ CD24^hi^ CD23^lo^ IgD^lo^ CD5^+^

The following Treg subpopulations were analyzed:tTregs (originating from CD4^+^ CD8^+^ thymic cells): CD3^+^ Foxp3^+^ CD4^+^ CD25^+^ GITR^+^ CD103^−^ GARP^−^ CD101^low^ CD304^high^pTregs (from conventional TCD4^+^ Foxp3^−^ cells): CD3^+^ Foxp3^+^ CD4^+^ CD25^+^ GITR^+^ CD103^−^ GARP^−^ CD101^lo^ CD304^lo^Activated Tregs: CD3^+^ Foxp3^+^ CD4^+^ CD25^+^ GITR^+^ CD304^+^, CD103^+^ CD101^hi^ GARP^+^. 

The percentage (mean) of viable T (CD3^+^) and B (CD19^+^) cells were:○blood: T cells—young mice 3.6%, old mice 2.4%; B cells—young mice 24.2%, old mice 25%.○lymph nodes: T cells—young mice 5.4%, old mice 4.7%; B cells—young mice 19.1%, old mice 21.1%.○spleen: T cells—young mice 2.7%, old mice 2.6%; B cells—young mice 47.7%, old mice 42.8%.

Before the analysis, the cells were washed with PBS containing 2% FBS (192× *g* at 4 °C). LSR Fortessa cytometer (BD Biosciences, San Jose, CA, USA) and FACSDiva V8.0.1 software (BD Biosciences, Franklin Lakes, NJ, USA) were used for analysis.

### 2.7. Statistical Analyses

Statistical analyses were carried out using GraphPad Prism 7 software. The normality of data distribution was determined using the Shapiro–Wilk data normality test. The statistical tests applied to individual data analyses are specified in figure footers; these were mostly based on the Mann–Whitney test. Differences between groups were considered to be statistically significant at *p* < 0.05.

## 3. Results

### 3.1. Effect of Calcitriol on Bregs in Blood, Spleen, and LNs

The levels of B10 cells (CD19^+^ CD21/CD35^+^ CD1d^+^ IgM^hi^ CD24^hi^ CD23^lo^ IgD^lo^ CD5^+^) remained similar in young and aged mice and were not significantly influenced by calcitriol (Figure 1A). Additionally, calcitriol did not influence the number of marginal zone B (MZB) cells (CD19^+^ CD21/CD35^+^ CD1d^+^) or transitional 2 MZ precursor B (T2-MZP) cells (CD19^+^ CD21/CD35^+^ CD1d^+^ IgM^hi^ CD24^hi^ CD23^hi^ IgD^hi^) in the blood; however, the aged mice possessed a lower percentage of MZB cells and a higher percentage of T2-MZP cells compared with the young mice (Figure 1A).

The percentages of B10 cells in LNs were similar in all the experimental mice groups. Aged control mice had a lower percentage of MZB cells, but their number slightly increased after calcitriol treatment. T2-MZP cells occurred at the same level in the LNs of young and aged control mice, but their levels were found to be higher in aged mice treated with calcitriol (Figure 1B). A higher percentage of CD19^+^ cells was observed in the LNs of aged mice compared with the LNs of young mice. However, calcitriol treatment significantly diminished this population in aged mice (Figure 1B).

Calcitriol did not exert any significant influence on the subsets of B cells in the spleens of both young and aged mice (Figure 1C). Higher percentages of MZB cells and lower percentages of T2-MZP cells were observed in both groups of aged mice (treated and control) compared with young mice; however, a lower percentage of B10 cells was observed in the old mice control group (Figure 1C). Figure 1D shows the gating scheme for individual cell populations exemplified by old calcitriol-treated mice.

### 3.2. Treg Subsets in Blood, LNs, and Spleen Cells of Young and Aged Mice Treated with Calcitriol

The levels of tTregs (originating from CD4^+^ CD8^+^ thymic cells; CD3^+^ Foxp3^+^ CD4^+^ CD25^+^ GITR^+^ CD103^−^ GARP^−^ CD101^lo^ CD304^hi^) and pTregs (from conventional T CD4^+^ Foxp3^−^ cells; CD3^+^ Foxp3^+^ CD4^+^ CD25^+^ GITR^+^ CD103^−^ GARP^−^ CD101^lo^ CD304^lo^) were similar in the blood of old and young mice. However, increased levels of pTregs were observed in calcitriol-treated old mice compared with young mice. A higher percentage of active Tregs (CD3^+^ Foxp3^+^ CD4^+^ CD25^+^ GITR^+^ CD304^+^ CD103^+^ CD101^hi^ GARP^+^) was observed in the blood of aged mice. Calcitriol did not significantly influence the Treg subsets in blood regardless of the age of the mice (Figure 2A). The only effect of calcitriol on Tregs in the bloodstream was a significantly decreased level of CD3^+^ Foxp3^+^ CD4^+^ CD25^+^ population in aged mice (Figure 3A). Similarly, Treg GITR^+^ (CD3^+^ Foxp3^+^ GITR^+^) occurred at a lower percentage in old mice treated with calcitriol (*p* = 0.0541; Figure 3A).

We observed lower percentages of Tregs (both CD3^+^ Foxp3^+^ and CD3^+^ Foxp3^+^ CD25^+^) (Figure 2B and Figure 3B, respectively) in the LNs of aged mice. Calcitriol significantly decreased the percentage of tTregs in the LNs of both young and aged mice (Figure 2B). Moreover, calcitriol significantly reduced the percentages of CD3^+^ Foxp3^+^ GITR^+^, CD3^+^ Foxp3^+^ CD101^lo^, CD3^+^ Foxp3^+^ CD103^+^, and CD3^+^ Foxp3^+^ CD25^+^ Tregs in young mice only (Figure 3B).

Calcitriol did not significantly influence the Treg subsets in the splenocytes of both young and aged mice (Figure 2C and Figure 3C). However, calcitriol treatment increased the percentage of pTregs in aged mice compared with young mice (Figure 2C). Figure 2D shows the gating scheme for individual cell populations exemplified by old calcitriol-treated mice.

### 3.3. Cytokine Levels in Plasma and Splenocyte Culture Supernatants

The plasma levels of the vitamin D_3_ metabolite 25(OH)D_3_ were comparable in all mice. Estradiol (E2) was not detected in the plasmas of old mice using the ELISA kit used, but it was detected in four of six young control mice and two of six young calcitriol-treated mice. Calcitriol did not significantly influence the level of either 25(OH)D_3_ or E2 (Figure 4A). The plasma level of immunoglobulin G (IgG) was significantly lower in aged mice than in young mice. The plasma levels of osteopontin (OPN) in both mice groups were reduced by calcitriol treatment; however, the differences were statistically significant only in the aged mice (Figure 4A).

IgG levels were significantly higher in the culture media of stimulated splenocytes of old mice compared with young mice treated with calcitriol. However, IgG levels did not differ in an age-dependent manner in control mice (Figure 4B). OPN levels were similar in young and aged mice, whereas IFN-γ and IL-10 were detectable only in aged mice (especially in the control group). On the other hand, TGF-β was detectable only in the splenocyte supernatants obtained from young control mice (Figure 4B).

### 3.4. Adipose Tissue from Young and Aged Mice Treated with Calcitriol

The level of 25(OH)D_3_ in adipose tissue was lower in aged mice, but only in the calcitriol-treated group. The levels of E2 and OPN were the same in young and aged mice, but calcitriol decreased their levels in aged mice (the difference was not significant). IgG levels were lower in adipose tissue of aged control mice but increased significantly in the calcitriol-treated group. The levels of IFN-γ, IL-10, and IL-35 in aged mice treated with calcitriol were diminished compared with their levels in young mice. Calcitriol significantly increased the level of IL-35 in young mice, whereas a decrease in IL-35 levels was observed in the adipose tissue of calcitriol-treated old mice. TGF-β levels were similar in all the groups of mice (Figure 4C).

### 3.5. Blood Morphological Parameters and Body Weights of Mice

Calcitriol did not affect the majority of blood morphological parameters (Figure 5A–D); however, it decreased the mean cell volume of erythrocytes in young mice (Figure 5D) and decreased the number and changed the characteristics (decreased plateletcrit and increased distribution width) of platelets in old mice (Figure 5E).

The body weights of young and old mice did not change during treatment (Figure 5F).

## 4. Discussion

One of the most noticeable changes that take place in the organs of the immune system during aging is a gradual decrease in the size and function of the thymus, which has an altered architecture [4,25]. Aging in mice is accompanied by decreased output of Tregs from the thymus and recirculation of activated Tregs from the periphery to the thymus where differentiation of Tregs is inhibited [26]. In our study, we observed that aged C57Bl/6/Foxp3^GFP^ mice were characterized by higher quantities of activated Tregs, despite lower levels of CD3^+^ Foxp3^+^ and CD3^+^ Foxp3^+^ CD4^+^ CD25^+^ populations. Additionally, Tregs in aged C57Bl/6/Foxp3^GFP^ mice seemed to exhibit enhanced immunosuppressive properties compared with young mice, with lower expression of GITR and higher expression of CD103 [27,28,29]. However, we observed higher percentages of pTregs in the blood and spleens of aged mice after treatment with calcitriol (compared with young treated mice). The pTregs generated throughout life mainly protect the organism from chronic inflammation and the semi-allogeneic fetus from rejection, whereas tTregs are produced at high levels in the first weeks of life, expand, and colonize the secondary lymphoid organs and peripheral tissues, protecting the organism from autoimmune diseases and promoting tissue repair [4]. Thus, taking into consideration that the incidence of low-grade chronic inflammation increases with age [30], it can be suggested that higher delivery of vitamin D may promote a desirable anti-inflammatory environment, which was reflected in our study by a higher proportion of pTreg subsets in aged mice treated with calcitriol. A scheme of differences between young and aged C57Bl/6/Foxp3^GFP^ mice is presented in Figure 6.

The most significant differences in age-related responses to calcitriol were observed in LNs. In both young and aged mice, calcitriol significantly decreased the percentage of tTregs in LNs. Interestingly, calcitriol also decreased the percentage of CD3^+^ Foxp3^+^ GITR^+^ cells, but only in the LNs of young mice. GITR (a member of the costimulatory TNF receptor superfamily) is a cell surface receptor that is highly expressed on Tregs. Its expression increases rapidly when activated by various stimuli. GITR stimulation allows the expansion of the CD8^+^ T effector memory cell population while inhibiting the suppressive properties of Tregs [27,28]. Moreover, we observed a significantly lower percentage of CD101^lo^ cells (among CD3^+^ Foxp3^+^) in the LNs of young mice. It was observed that the proliferation of T cells in vitro, driven by an alloantigen, was highly suppressed by CD101^hi^ Tregs compared to CD101^lo^ Tregs [31]. On the other hand, CD103^+^ Tregs, the most potent suppressors of inflammatory processes in mice [29], were significantly diminished in the LNs of young mice treated with calcitriol but not in the LNs of old mice treated with calcitriol. Therefore, a lower percentage of tTregs and decreased proportions of GITR^+^, CD101^lo^, and CD103^+^ populations may, in combination, be considered as evidence of the calcitriol modulatory action toward Tregs in healthy young mice, which is directed in general by an increase in the suppressive potential of Tregs, despite their lower numbers.

B lymphopoiesis declines with age, and bone marrow tissue is replaced by adipose tissue. Moreover, adipose tissue cytokines are known for inhibiting B lymphopoiesis [32,33]. MDSCs, a population that is documented to expand with aging, are also involved in this process [34]. MDSCs can also induce the expansion of Bregs in mice [35]. We observed a higher level of CD19^+^ cells in LNs of aged C57Bl/6/Foxp3^GFP^ mice. The subpopulations of CD19^+^ cells, defined as Bregs, differed depending on the tissue analyzed. We observed lower levels of B10 and T2-MZP cells but a higher level of MZB cells in the spleens of aged mice compared with those of young mice. MZB cells are among the first population of cells encountered by blood-borne antigens and are presumed to have a critical role in thymic-independent host defense against bacterial pathogens [36]. The functioning and migration of splenic MZB cells were reported to be impaired in elderly mice, and aged MZB cells were retained within the marginal zone in the spleen [37]. In accordance with these observations, the levels of MZB cells in the blood were lower while the numbers of T2-MZP cells (“younger”, precursor cells [38]) were higher in aged compared with young C57Bl/6/Foxp3^GFP^ mice. 

Bregs were not significantly affected by calcitriol. We observed decreased percentages of CD19^+^ B lymphocytes in the LNs of aged mice. Wang et al. have previously observed decreased levels of IL-10 mRNA and protein in B lymphocytes isolated from the peripheral blood of 35–55-year-old patients after ex vivo treatment with calcitriol or its analog EB1089 [21]. However, in the present study, we detected IL-10 only in fatty tissue, where its level was lowest in calcitriol-treated old mice, in accordance with the results of the study by Wang et al. We observed increased IL-10 levels in tumor tissue of young mice following treatment with calcitriol in our previous study conducted in mice bearing 4T1 mammary gland tumors [22], whereas no such effect was noted in old mice [39]. This correlates with the increased immune response of Tregs observed in young mice [22] and the opposite effect observed in these cells in old mice [18]. The impact of calcitriol on IL-10 secretion varied among the cells studied or tissues analyzed or based on the disease investigated. As mentioned, Wang et al. showed decreased levels of IL-10 in human B lymphocytes [21], whereas Heine et al. showed increased expression of IL-10 in human B cells treated with calcitriol ex vivo [40]. However, the first study was conducted on patients with pituitary tumors, whereas in the second one B cells from healthy volunteers were analyzed. A limitation of our current study conducted on young and old C57BL/6/Foxp3^GFP^ mice is the lack of examination of ex vivo IL-10 secretion or its intracellular staining in separated B10 cells, which requires further research. Therefore, to fully explain the age-dependent impact of calcitriol on the production of this cytokine by various cells in healthy and affected (e.g., tumor-bearing) organisms, additional studies are needed.

Calcitriol did not affect IgG levels in plasma and splenocyte supernatants, but increased IgG levels in the fatty tissue of old mice; the opposite effect was observed in young mice. It has been documented that B lymphocytes can accumulate in the adipose tissue and produce IgG [41]. Tregs can also accumulate in this tissue and exhibit different characteristics from the Tregs present in other tissues [42]. Thus, we can speculate that in adipose tissue, calcitriol exerted contrasting effects on these cells in young and aged mice. This was confirmed by the finding that calcitriol increased IL-35 levels in young mice, whereas the levels of this cytokine were decreased in aged mice. Taking into consideration that IL-35 promotes the development of Tregs and Bregs [43,44], it is worth emphasizing that calcitriol seemed to enhance the immunosuppressive characteristics of adipose tissue in young mice while inhibiting such properties in aged mice. This conclusion is also supported by the low levels of IL-10 and IFN-γ observed in calcitriol-treated old mice. IL-35 requires additional immune-regulatory cytokines, such as IL-10 and TGF-β, to induce maximal anti-inflammatory effects [43]. Given that increased production of proinflammatory cytokines such as IL-6, TNF-α, and IFN-γ by B cells was reported in obese individuals [45,46], the effect of calcitriol observed in aged mice may have contributed to the normalization of homeostasis in adipose tissue. 

OPN levels in plasma and adipose tissue of aged mice were diminished by calcitriol treatment; however, OPN levels were increased in splenocyte supernatants from young mice. Calcitriol is known to stimulate various cells to secrete OPN via a vitamin D-response element in the promoter region of the OPN gene (*Spp1*) [47,48,49]. At the same time, OPN stimulates tumor growth, metastasis, and angiogenesis [50,51,52]. However, in our previous studies, it turned out that stimulated splenocytes isolated from calcitriol- or PRI-2191 (calcitriol analog)-treated young mice bearing 4T1 mammary gland cancer had lower levels of OPN compared with control mice, particularly during the early stages of tumor progression [22]. We did not find any other studies mentioning the inhibitory effect of calcitriol on OPN secretion. A possible explanation for this finding may be the decrease in the number of activated T and B lymphocytes following calcitriol treatment, which has been previously described [12,53], and that only activated T and B lymphocytes produce OPN [54]. In the present study, calcitriol showed a tendency to decrease the percentage of active Tregs in the blood and LNs and to increase their percentage in splenocytes. Such effects are compatible with decreased levels of OPN in the plasma and increased levels in splenocyte culture. The elevated level of OPN observed in the tumor tissue of young mice after treatment with calcitriol and its analogs in our previous study [19] might be a result of fibroblast stimulation, which is the major source of OPN in tumor tissue [55,56]. On the other hand, OPN levels decreased in plasma and tumor tissue in old mice bearing 4T1 following calcitriol treatment [20]. Further research is needed to clarify the influence of calcitriol on OPN secretion by immune system cells in tumor-bearing and healthy young vs. old organisms.

## 5. Conclusions

Calcitriol treatment had different effects on young and old female mice, mainly on Treg phenotypes and cytokine secretion (Figure 7). In young mice, calcitriol decreased the expression of GITR and CD103 on Tregs, diminished the percentages of CD3+ Foxp3+ CD101^lo^ cells and tTregs in LNs, and increased the IL-35 levels in adipose tissue. Similarly, calcitriol decreased the percentage of tTregs in aged mice; however, OPN levels in the plasma decreased whereas IgG levels in the adipose tissue increased in aged mice following calcitriol treatment. Furthermore, IFN-γ, IL-10, and IL-35 levels were lowest in the adipose tissue of the calcitriol-treated aged mice.

In general, the action of calcitriol in healthy young mice is associated with enhanced suppressive potential of regulatory cells (mostly observable in LNs and adipose tissue). On the other hand, the expressions of cytokines in adipose tissue and OPN in the plasma contribute to decreased activation of Tregs and Bregs in calcitriol-treated aged mice.

## Figures and Tables

**Figure 1 nutrients-16-00049-f001:**
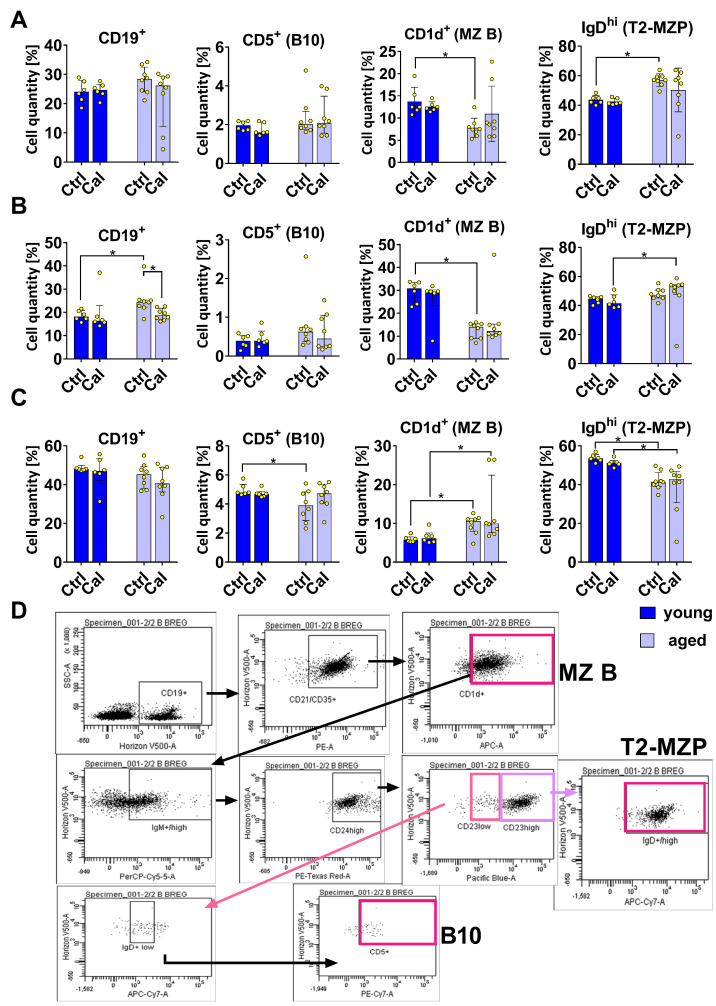
Phenotypes of B cells from young and aged mice treated with calcitriol. (**A**) Blood. (**B**) Lymph Nodes. (**C**) Spleen. (**D**) Example dot plots: blood cells from aged mice treated with calcitriol are labeled. Cells were incubated with antibodies labeled for multicolor flow cytometry analysis (isotype controls were also included). The following B cell subpopulations were analyzed: MZB cells: CD19^+^ CD21/CD35^+^ CD1d^+^; T2-MZP cells: CD19^+^ CD21/CD35^+^ CD1d^+^ IgM^hi^ CD24^hi^ CD23^hi^ IgD^hi^; B10 cells: CD19^+^ CD21/CD35^+^ CD1d^+^ IgM^hi^ CD24^hi^ CD23^lo^ IgD^lo^ CD5^+^. Number of analyses performed: 6–10. Statistical analysis: Mann–Whitney test; * *p* < 0.05.

**Figure 2 nutrients-16-00049-f002:**
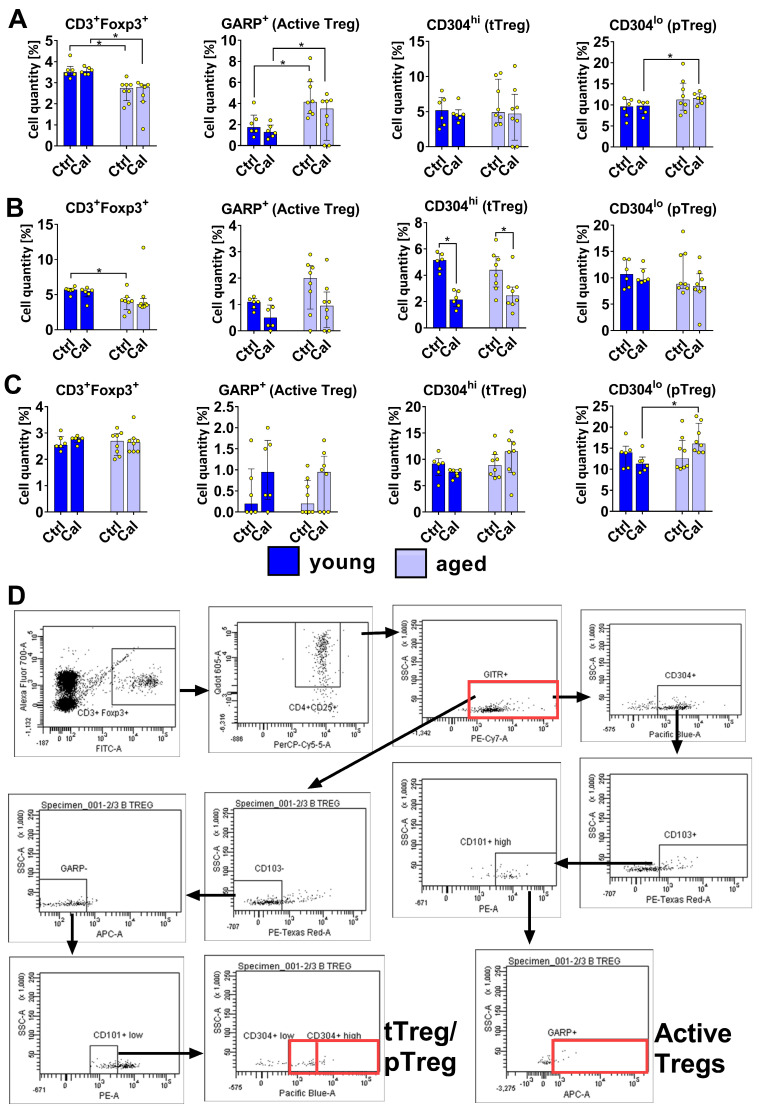
Phenotypes of Tregs from young and aged mice treated with calcitriol. (**A**) Blood. (**B**) Lymph Nodes. (**C**) Spleen. (**D**) A gating illustration of the analysis of Tregs in LNs obtained from aged mice treated with calcitriol is shown as an example. Cells were incubated with antibodies labeled for multicolor flow cytometry analysis (isotype controls were included as well). The following Treg subpopulations were analyzed: tTregs: CD3^+^ Foxp3^+^ CD4^+^ CD25^+^ GITR^+^ CD103^−^ GARP^−^ CD101^low^ CD304^high^; pTregs: CD3^+^ Foxp3^+^ CD4^+^ CD25^+^ GITR^+^ CD103^−^ GARP^−^ CD101^lo^ CD304^lo^; activated Tregs: CD3^+^ Foxp3^+^ CD4^+^ CD25^+^ GITR^+^ CD304^+^, CD103^+^ CD101^hi^ GARP. Dark blue indicates young mice and light blue indicates aged mice. Number of analyses performed: 6–8. Statistical analysis: Mann–Whitney test; * *p* < 0.05.

**Figure 3 nutrients-16-00049-f003:**
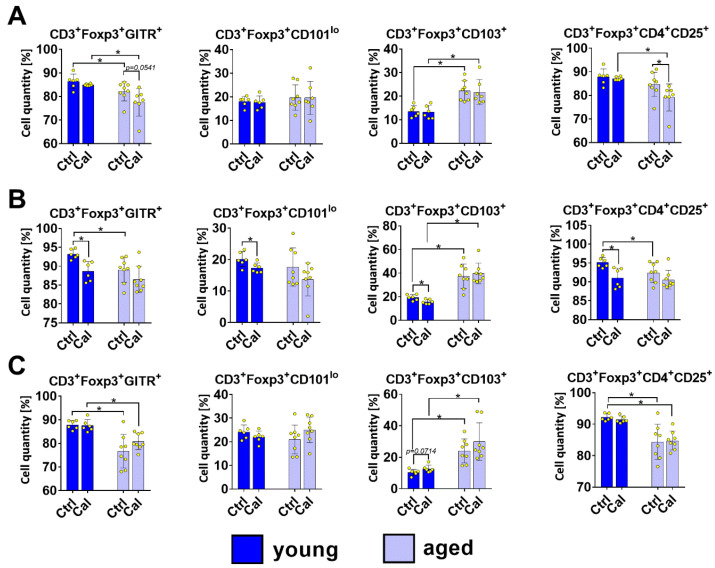
Different populations of Tregs in young and aged mice treated with calcitriol. Treg subset analysis in (**A**) blood, (**B**) LNs, and (**C**) spleen. Cells were incubated with antibodies labeled for multicolor flow cytometry analysis (isotype controls were included). The following Treg subpopulations were analyzed: CD3^+^ Foxp3^+^ GITR^+^; CD3^+^ Foxp3^+^ CD101^low^; CD3^+^ Foxp3^+^ CD103^+^; CD3^+^ Foxp3^+^ CD4^+^ CD25^+^. Dark blue indicates young mice and light blue indicates aged mice. Number of analyses performed: 6–8. Statistical analysis: Mann–Whitney test; * *p* < 0.05.

**Figure 4 nutrients-16-00049-f004:**
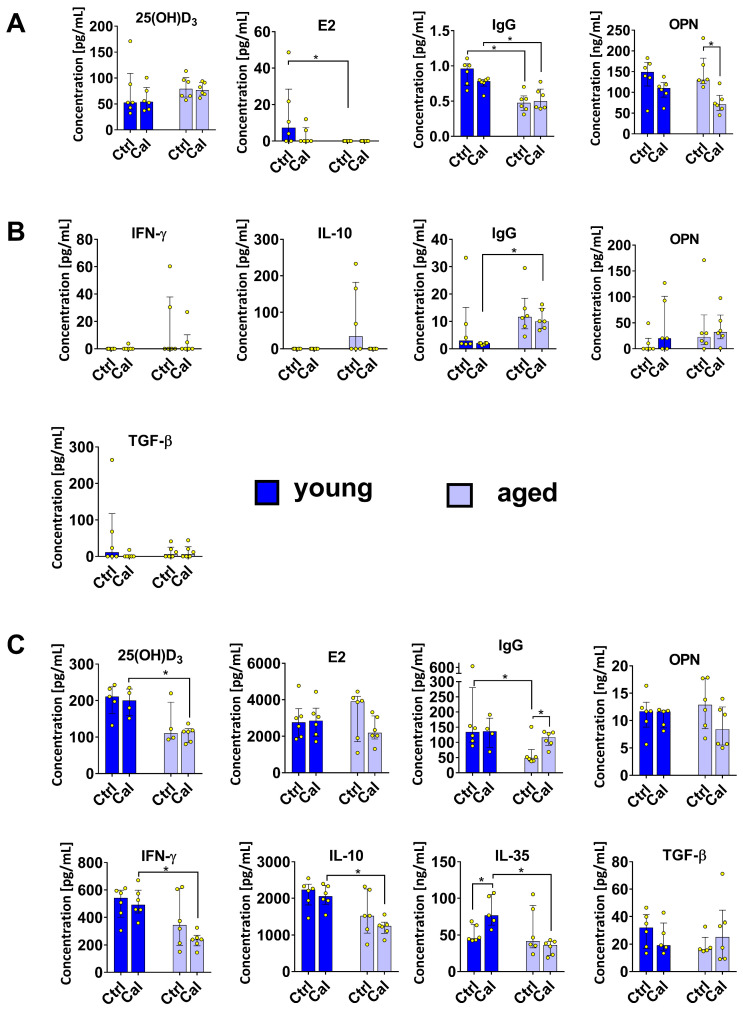
Level of vitamin D_3_ metabolite, E2, IgG, and selected cytokines in the plasma, splenocyte culture supernatants, and adipose tissue of young and aged mice treated with calcitriol (median ± IQR (interquartile range)). (**A**) Plasma levels of vitamin D_3_ metabolite 25(OH)D_3_, E2, IgG, and OPN. (**B**) Splenocytes were incubated with PMA for 4 h, and supernatants were then collected and analyzed. The levels of IFN-γ, IL-10, IgG, OPN, and TGF-β in the supernatants were determined. (**C**) Levels of 25(OH)D_3_, E2, IgG, OPN, IFN-γ, IL-10, IL-35, and TGF-β in adipose tissue. Dark blue indicates young mice and light blue indicates aged mice. Number of analyses performed: 5–6. Statistical analysis: Mann–Whitney test; * *p* < 0.05.

**Figure 5 nutrients-16-00049-f005:**
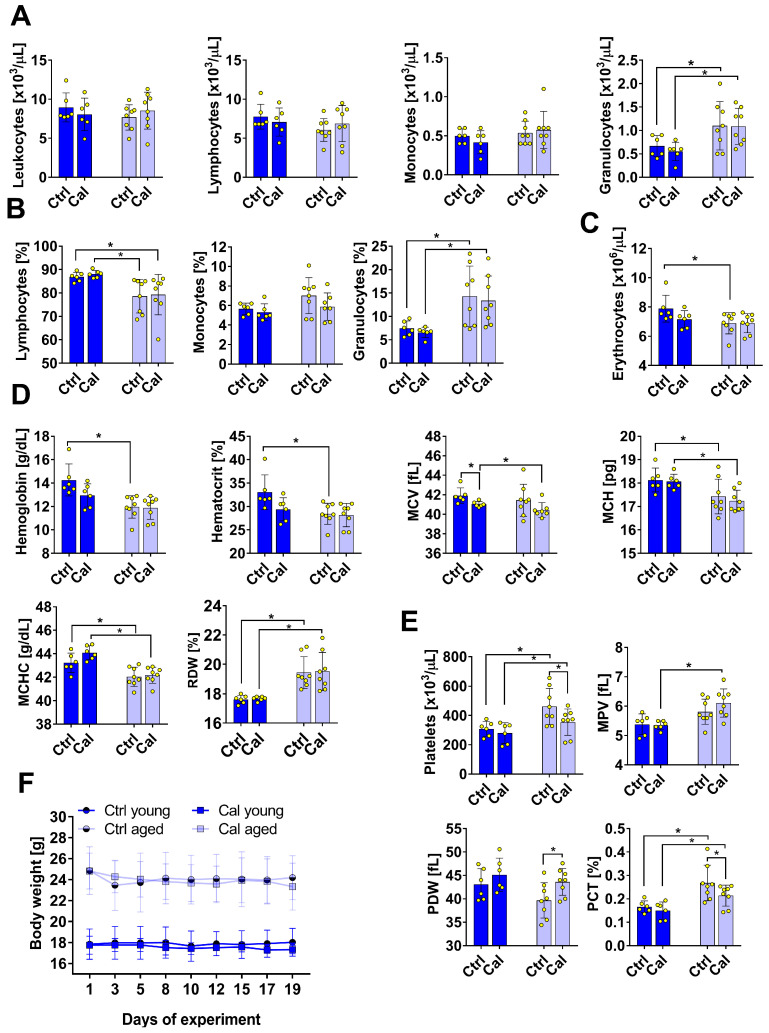
Blood morphological parameters and body weights of young and aged mice treated with calcitriol. (**A**) Numbers of leukocytes, lymphocytes, monocytes, and granulocytes. (**B**) Percentages of lymphocytes, monocytes, and granulocytes. (**C**) Number of erythrocytes. (**D**) Hemoglobin (Hb), hematocrit, mean cell volume (MCV), red cell distribution width (RDW), mean corpuscular hemoglobin concentration (MCHC), and mean corpuscular hemoglobin (MCH). (**E**) Platelet number, plateletcrit (PCT), platelet distribution width (PDW), and mean platelet volume (MPV). (**F**) Variations in body weight throughout treatment. Dark blue indicates young mice and light blue indicates aged mice. Number of analyses performed: 5–6. Statistical analysis: Mann–Whitney test; * *p* < 0.05.

**Figure 6 nutrients-16-00049-f006:**
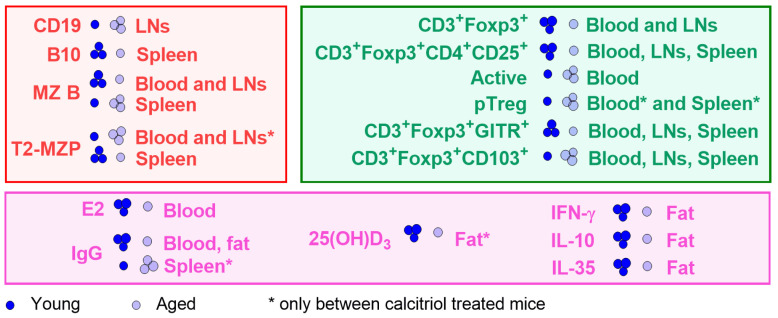
Summary of differences between young and aged C57Bl/6/Foxp3^GFP^ mice used as models in our study. Red, Bregs phenotype; green, Tregs phenotype; pink, levels of E2 (17β-estradiol), IgG, 25(OH)D_3_, and cytokines in different tissues. One and three circles indicate the lower and higher levels of the marker being tested, respectively.

**Figure 7 nutrients-16-00049-f007:**
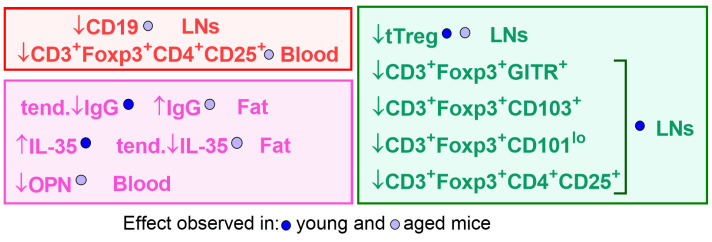
Summary of the effects of calcitriol treatment on the immune response of young and aged C57Bl/6/Foxp3^GFP^ mice. Red, Bregs phenotype; green, Tregs phenotype; pink, levels of E2 (17β-estradiol), IgG, 25(OH)D_3_, and cytokines. The arrows indicate low and high levels of the marker being tested in calcitriol-treated mice compared with control mice.

**Table 1 nutrients-16-00049-t001:** List of antibodies used for flow cytometry analysis.

Determined Protein	Mark	Clone	Manufacturer
CD19	Brilliant Violet 510	6D5	Biolegend, San Diego, CA, USA
CD21/CD35	PE	7E9	
CD1d (CD1.1, Ly-38)	Alexa Fluor 647	1B1	
IgM	PerCP/Cy5.5	RMM-1	
IgD	APC/Fire 750	11-26c.2a	
CD23	Brilliant Violet 421	B3B4	
CD24	PE/Dazzle 594	M1/69	
CD5	PE/Cy7	53-7.3	
CD3	Alexa Fluor 700	17A2	
CD4	PerCP/Cy5.5	GK1.5	
CD25	Brilliant Violet 605	PC61	
CD357 (GITR)	PE/Cy7	DTA-1	
CD101	PE	Moushi101	Invitrogen, Carlsbad, CA, USA
CD103	PE/Dazzle 594	2E7	Biolegend, San Diego, CA, USA
CD304 (neuropilin-1)	Brilliant Violet 421	3E12	
GARP	Alexa Fluor 647	F011-5	

## Data Availability

All data generated or analyzed during this study are included in this published article.

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
