# Peer review of "Age-Dependent Effect of Calcitriol on Mouse Regulatory T and B Lymphocytes"

_nutrients, 2023, doi:10.3390/nu16010049_

Round 1
Reviewer 1 Report
Comments and Suggestions for Authors
The study"Age-dependent effect of calcitriol on mouse regulatory T and B 2 lymphocytes" investigates the age-dependent effects of calcitriol, an active form of vitamin D3, on the immune system, focusing on regulatory T (Treg) and B (Breg) lymphocytes in healthy young and old female mice. The authors observe distinct alterations in immune cell populations and cytokine profiles in response to calcitriol treatment in young versus aged mice.
The proposed manuscript could be interesting for the relevant specialists, since immunomodulatory role of calcitriol, age-related changes in immune cells, differential effects of calcitriol in young vs. aged mice, tissue-specific responses has been demontrated.
However, despite in general well-performed study, there are some important shortcommings regarding represented results.
Critical remarks from the statistical point of view.
The results presented in Figure1B, Figure2A,B,C, Figure4A,B,C are mostly inconsistent. Despite the fact that authors correctly described in paragraph 2.7 (Statistical analysis) the procedure of Shapiro-Wilk data normality proof, it seems that not all data fit to normal distribution. As a consequence of the obviously weak basic knowledge about data distribution and correct evaluation of the central trend, mostly meaningless results are presented in these Figures, where dispersions in many cases are close or higher than the mean values (Figure1B; Figure2A,B,C; Figure4A,B) and the lower limit of 1SD goes in minus (not taking into account also 2SD and 3SD, see below rules of the distribution, Fig.2). Therefore that is completely non-logical and incorrect calculation since the data analyzed belongs to ratio scale, and are continuous with real null value and these values cannot be with minus sign. Instead of mean, the median with IQR must be used.
Fig.2
Conclusion: Overall, the research underscores the age-dependent effects of calcitriol on immune cell populations and cytokine production in healthy mice. The findings suggest that calcitriol exerts differential immunomodulatory actions based on age, emphasizing the complexity of immune regulation. This highlights the potential for targeted immune interventions considering age-related differences in response to calcitriol.
The Manuscript must be revised according critical remarks.
Author Response
Dear Reviewer,
thank you very much for all your valuable comments. All answers to your suggestions and questions are placed below your comments in blue fonts. Thank you for reviewing this manuscript.
Reviewer 1
Comments and Suggestions for Authors
The study "Age-dependent effect of calcitriol on mouse regulatory T and B 2 lymphocytes" investigates the age-dependent effects of calcitriol, an active form of vitamin D3, on the immune system, focusing on regulatory T (Treg) and B (Breg) lymphocytes in healthy young and old female mice. The authors observe distinct alterations in immune cell populations and cytokine profiles in response to calcitriol treatment in young versus aged mice.
The proposed manuscript could be interesting for the relevant specialists, since immunomodulatory role of calcitriol, age-related changes in immune cells, differential effects of calcitriol in young vs. aged mice, tissue-specific responses has been demontrated.
However, despite in general well-performed study, there are some important shortcommings regarding represented results.
Critical remarks from the statistical point of view.
The results presented in Figure1B, Figure2A,B,C, Figure4A,B,C are mostly inconsistent. Despite the fact that authors correctly described in paragraph 2.7 (Statistical analysis) the procedure of Shapiro-Wilk data normality proof, it seems that not all data fit to normal distribution. As a consequence of the obviously weak basic knowledge about data distribution and correct evaluation of the central trend, mostly meaningless results are presented in these Figures, where dispersions in many cases are close or higher than the mean values (Figure1B; Figure2A,B,C; Figure4A,B) and the lower limit of 1SD goes in minus (not taking into account also 2SD and 3SD, see below rules of the distribution, Fig.2). Therefore that is completely non-logical and incorrect calculation since the data analyzed belongs to ratio scale, and are continuous with real null value and these values cannot be with minus sign. Instead of mean, the median with IQR must be used.
Thank you very much for your valuable comments.
- The inconsistency in our data presented on Figure 4 comes from the lack of splenocyte probes we would need to perform all analyses. We tried to focus on the most important cytokines produced by splenocytes that’s why in Fig 4B the INFÉ£, IL-10, IgG, OPN and TGF-β results are presented, without 25(OH)D3 and E2.
- As it was mentioned in the description of each figure the most of the data was analysed by the nonparametric Mann-Whitney U-test. The information is now added also to point 2.7.
- As you suggested, I have changed the figures (Fig 1, Fig 2, Fig 4) in the manuscript. They are presenting now median with interquartile range. Thank you for your clear perspective and critical comments, which have helped to improve the clarity of the presented data.

Reviewer 2 Report
Comments and Suggestions for Authors
Sniezewska et al. report on calcitriol affect on Treg and Breg cells based on age.
Abstract; recommend no abbreviations in abstracts; authors need to remember first usage of abbreviations requires full spelling LN, OPN, etc
Introduction:
“with increasing age, naïve CD4+ CD45RA+ cells are replaced by memory CD4+ CD45RA- cells.” (Reference ?) T CD8+ should be CD8+ T-lymphocytes
Results and Discussion:
The study was undertaken to assess the influence of calcitriol on immune populations of mice not with cancer or other immune modulators as had been done in prior published reports. “Old” mice are usually considered to be specified pathogen free (SPF) mice of about 2 years of age; whereas, the “old” mice used in this report were only 1 year of age. Some comment should be noted for this difference, and it should be indicated if the mice were maintained with SPF conditions and if background exposures to stimulants such as endotoxin in food or bedding were evaluated. Similarly, the source of the calcitriol needs to be provided and if it was free of endotoxin.
The emphasis was on Treg and B reg cells; however, the effects of age and calcitriol should be provided for all T (CD4 and CD8) and B cells so that the proportions of the Treg and Breg cells can be better related to the overall immune system effects of calcitriol with age. For the flow cytometry, it would be important to show the dot plots and percentages of initial light scatter populations and viability of all T and B cells of the young and old untreated and treated mice before showing the subpopulations. There should have been analysis (percentages of viable total numbers of T (CD3+) and B (CD19+) cells in each site as well as the subpopulations. Although cell viability was assayed, there was no indication in methods if the cytometric analyses excluded non-viable cells or cell aggregates. For better evaluation of cytokines such as IFN-γ and IL-10 knowing the T cell percentages including of CD45RA- (or CD45RO+) memory cells would be important. Similarly, for the B cell populations and IgG levels it would have been important to have had the plasma and culture supernatant levels for IL-6 as well as other cytokines know to affect B cells such as IL-4 and IL-5. To be able to conclude the immunosuppressive differences among the groups is based on the cell proportions of the assayed subpopulations, the analysis should have included adding the old vs young cells into the cultures of the young vs old cells to assay influences; this could have been done since the cells are syngeneic and would have been more direct assessment of suppressive function.
The in vivo differences of age and calcitriol on IgG vs the increased IgG level ex vivo (Figure 4) would have been more understandable if IgG expressing B cells were included for analysis to differentiate cells able to synthesize IgG or synthesis/B cells. As stated before, the level of T cell cytokines also would help to differentiate effects.
The proportions of the leukocytes is not discussed with regard to other factors known to change with age such as redox with more reactive oxygen species with age. Additionally, there is known to be more oxidative stress with an increased neutrophil to lymphocyte ratio (NLR) and as shown in Figure 5 the older mice had more granulocytes and few lymphocytes suggesting a higher NLR.
The study shows some interesting Treg and Breg subset differences however effects of age and calcitriol would have been made better with inclusion of better cytometric analysis of the T and B cells including proportions of cells making some of the known regulatory cytokines such as IFN- γ, IL-10, TGF-β, and IL-6 and IL-17 which were not measured.
Comments on the Quality of English LanguageEnglish quality is good
Author Response
Dear Reviewer,
thank you very much for all your valuable comments. All answers to your suggestions and questions are placed below your comments in blue fonts. Thank you for reviewing this manuscript.
Reviewer 2
Comments and Suggestions for Authors
Sniezewska et al. report on calcitriol affect on Treg and Breg cells based on age.
- Abstract; recommend no abbreviations in abstracts; authors need to remember first usage of abbreviations requires full spelling LN, OPN, etc
- The abbreviations were deleted from the abstract and corrections are made.
- Introduction:
“with increasing age, naïve CD4+ CD45RA+ cells are replaced by memory CD4+ CD45RA- cells.” T CD8+ should be CD8+ T-lymphocytes
It is the older one Reference, it is No 2 - Kudlacek, S., Jahandideh-Kazempour, S., Graninger, W., Willvonseder, R. & Pietschmann, P. Differential Expression of Various T Cell Surface Markers in Young and Elderly Subjects. Immunobiology 192, 198–204 (1995).
- Results and Discussion:
The study was undertaken to assess the influence of calcitriol on immune populations of mice not with cancer or other immune modulators as had been done in prior published reports. “Old” mice are usually considered to be specified pathogen free (SPF) mice of about 2 years of age; whereas, the “old” mice used in this report were only 1 year of age. Some comment should be noted for this difference, and it should be indicated if the mice were maintained with SPF conditions and if background exposures to stimulants such as endotoxin in food or bedding were evaluated. Similarly, the source of the calcitriol needs to be provided and if it was free of endotoxin.
- In this study our goal was to evaluate the differences in immune response to calcitriol supplementation in healthy young (premenopausal, fertile) mice versus healthy aged (postmenopausal) mice. Our previous findings on mammary cancer bearing mice revealed some differences according VDR and osteopontin genes expression and IL-17 levels in Th17 cells between young and old ovariectomized mice
(Anisiewicz A, Pawlik A, Filip-Psurska B, Wietrzyk J., Differential Impact of Calcitriol and Its Analogs on Tumor Stroma in Young and Aged Ovariectomized Mice Bearing 4T1 Mammary Gland Cancer., Int J Mol, Sci. 2020 Sep 2;21(17):6359. doi: 10.3390/ijms21176359.PMID: 32887237.,
Pawlik A, Anisiewicz A, Filip-Psurska B, Klopotowska D, Maciejewska M, Mazur A, Wietrzyk J, Divergent Effect of Tacalcitol (PRI-2191) on Th17 Cells in 4T1 Tumor Bearing Young and Old Ovariectomized Mice.,.Aging Dis. 2020 Mar 9;11(2):241-253. doi: 10.14336/AD.2019.0618. eCollection 2020 Apr.PMID: 32257539.,
Anisiewicz A, Filip-Psurska B, Pawlik A, Nasulewicz-Goldeman A, Piasecki T, Kowalski K, Maciejewska M, Jarosz J, Banach J, Papiernik D, Mazur A, Kutner A, Maier JA, Wietrzyk J., Calcitriol Analogues Decrease Lung Metastasis but Impair Bone Metabolism in Aged Ovariectomized Mice Bearing 4T1 Mammary Gland Tumours.; Aging and Disease, Volume 10, Number 5; 977-991, October 2019)
- As mice older than 30-40 weeks were considered as aged animals (biochemically, postmenopausal) in this study we have taken 1 year old (52 weeks) C57BL/J6FoxP3 female mice and young (6 weeks) mice. Old mice were our postmenopausal healthy model and young mice are the premenopausal one. All mice where maintained in SPF conditions in the animal facility, what means it is a pathogen free zone in the animal facility. This mice never entered the conventional conditions area. All mice in this experiment were feed with a special prepared feed by the manufacturer (free from endotoxins). The calcitriol administered to the mice was also prepared according to the GMP methods, in accordance with pharmaceutical guidelines prepared by the Cayman Chemicals. The information is now added to the Materials and Methods section.
The emphasis was on Treg and B reg cells; however, the effects of age and calcitriol should be provided for all T (CD4 and CD8) and B cells so that the proportions of the Treg and Breg cells can be better related to the overall immune system effects of calcitriol with age. For the flow cytometry, it would be important to show the dot plots and percentages of initial light scatter populations and viability of all T and B cells of the young and old untreated and treated mice before showing the subpopulations.
- In my opinion It would complicate the whole picture far more than it is.
There should have been analysis (percentages of viable total numbers of T (CD3+) and B (CD19+) cells in each site as well as the subpopulations. Although cell viability was assayed, there was no indication in methods if the cytometric analyses excluded non-viable cells or cell aggregates.
- Yes, the non-viable (DAPI staining) cells and aggregates were excluded from the analysis.
- The percent (mean) of viable T (CD 3+) and B (CD19+) cells were:
- in blood: T cells – young mice 6%, old mice 2.4%, B cells - young mice 24.2%, old mice 25%
- lymph nodes: T cells - young mice 4%, old mice 4.7%, B cells - young mice 19.1%, old mice 21.1%
- spleen: T cells - young mice 2.7%, old mice 2.6%, B cells- young mice 47.7%, old mice 42.8%
For better evaluation of cytokines such as IFN-γ and IL-10 knowing the T cell percentages including of CD45RA- (or CD45RO+) memory cells would be important. Similarly, for the B cell populations and IgG levels it would have been important to have had the plasma and culture supernatant levels for IL-6 as well as other cytokines know to affect B cells such as IL-4 and IL-5.
- We didn’t have enough samples for all this cytokine evaluations. But we will take it into consideration in our future plans. Thank you for the valuable tip.
To be able to conclude the immunosuppressive differences among the groups is based on the cell proportions of the assayed subpopulations, the analysis should have included adding the old vs young cells into the cultures of the young vs old cells to assay influences; this could have been done since the cells are syngeneic and would have been more direct assessment of suppressive function.
- If I understood properly, we don’t want to evaluate the cross influence by culturing old and young cells together. The goal was to evaluate the immunological background of calcitriol administrations as supplementation in healthy young and old, postmenopausal animals (similarly like it is evaluated in some human studies).
The in vivo differences of age and calcitriol on IgG vs the increased IgG level ex vivo (Figure 4) would have been more understandable if IgG expressing B cells were included for analysis to differentiate cells able to synthesize IgG or synthesis/B cells. As stated before, the level of T cell cytokines also would help to differentiate effects.
- We didn’t have enough samples for all this cytokine evaluations. But we will take it into consideration in our future plans. Thank you for the valuable tip.
The proportions of the leukocytes is not discussed with regard to other factors known to change with age such as redox with more reactive oxygen species with age. Additionally, there is known to be more oxidative stress with an increased neutrophil to lymphocyte ratio (NLR) and as shown in Figure 5 the older mice had more granulocytes and few lymphocytes suggesting a higher NLR.
- We didn’t have enough samples for all this studies. But we will take it into consideration in our future plans. Thank you for the valuable tip.
The study shows some interesting Treg and Breg subset differences however effects of age and calcitriol would have been made better with inclusion of better cytometric analysis of the T and B cells including proportions of cells making some of the known regulatory cytokines such as IFN- γ, IL-10, TGF-β, and IL-6 and IL-17 which were not measured.
IL-6 and IL-17 were not measured in this study. IL-17 levels were analysed in cancer bearing mice (some studies published but also unpublished data). IFN- γ, IL-10, TGF-β were measured We will take it also into account in future studies.
Reviewer 3 Report
Comments and Suggestions for Authors
The authors use female C57Bl/6/Foxp3gfp mice to compare the effect of treatment with calcitriol on immune cells of older mice (52-weeks of age) with those of young mice (6 weeks of age). The results are interesting, revealing both that there are differences in Treg cells and B cells when comparing 6 month old mice to 52-week old mice and that they groups vary in their response to calcitriol. The authors conclude that the action of calcitriol in healthy young mice is in general associated with the enhanced suppressive potential of regulatory cells. On the other hand, the expression of cytokines in adipose tissue and OPN in the plasma contributed to the decreased activation of Tregs and Bregs in calcitriol treated aged mice. These observations are important and have implications for treatment of patients in the clinics.
However, there are some points that need to be clarified
1. Only one dose of calcitriol is used. The differences noted in the manuscript would be more convincing if several doses of calcitriol were used (a dose response).
2. Similarly, only two age points were used. The differences would be more convincing if an age group in between was used (for example, 3 month old mice).
3. The manuscript is not easy to read, partly because the authors do not provide a clear explanation of the varying types of suppressive cells, suppressive cell markers, and suppressive cytokines. A clear explanation of the cells being studied and why they were selected for study, as well as a better explanation of what their increase or decrease is expected to cause in the mouse would enhance the clarity of this manuscript.
Author Response
Dear Reviewer,
thank you very much for all your valuable comments. All answers to your suggestions and questions are placed below your comments in blue fonts. Thank you for reviewing this manuscript.
Reviewer 3
Comments and Suggestions for Authors
The authors use female C57Bl/6/Foxp3gfp mice to compare the effect of treatment with calcitriol on immune cells of older mice (52-weeks of age) with those of young mice (6 weeks of age). The results are interesting, revealing both that there are differences in Treg cells and B cells when comparing 6 month old mice to 52-week old mice and that they groups vary in their response to calcitriol. The authors conclude that the action of calcitriol in healthy young mice is in general associated with the enhanced suppressive potential of regulatory cells. On the other hand, the expression of cytokines in adipose tissue and OPN in the plasma contributed to the decreased activation of Tregs and Bregs in calcitriol treated aged mice. These observations are important and have implications for treatment of patients in the clinics.
However, there are some points that need to be clarified
- Only one dose of calcitriol is used. The differences noted in the manuscript would be more convincing if several doses of calcitriol were used (a dose response).
- Only one, biologically active and safe dose was used. The dose was achieved in many experiments performed before (for example: Wietrzyk et al, 2007 Anticancer Res 2007 Sep-Oct;27(5A):3387-98., AA, BFP, AP et all. 2019, doi:10.14336/AD.2018.0921). Lover doses of calcitriol could be ineffective and higher doses reveal hazardous calcemic toxicity to the animals, what was proven before (Wietrzyk et all, 2005, 2007, 2008 and more, Trynda et all, 2015, Milczarek et all, 2015).
- Similarly, only two age points were used. The differences would be more convincing if an age group in between was used (for example, 3 month old mice).
- Mice older than 40 weeks are recognized as old mice (literature,). There was no ethical reason to include in the experiment a group of mice in the middle (for example 3 months old) between the young, fertile, and old mice. Both groups imitate the pre- and postmenopausal status, what was of our interest since we found out some significant immunological differences between young and old (pre- ans postmenopausal models) mice bearing mammary cancer. Calcitriol (or its analog) treatment in both tumor bearing groups revealed different metastatic potential of the implanted cancer. In this study healthy mice were taken to observe the immunological background after calcitriol administration in comparison the non-supplemented control group.
- The manuscript is not easy to read, partly because the authors do not provide a clear explanation of the varying types of suppressive cells, suppressive cell markers, and suppressive cytokines. A clear explanation of the cells being studied and why they were selected for study, as well as a better explanation of what their increase or decrease is expected to cause in the mouse would enhance the clarity of this manuscript.
- It is not easy to predict what results should we expect in this mouse strain because of the fact that ee used a modified version of the classic C57Bl/6 strain. It’s a C57Bl/6/Foxp3GFP Obviously, they are still C57Bl/6 mice, but the mice express green fluorescent protein (GFP) under the control of the Foxp3 gene promoter to evaluate the effect of calcitriol on Tregs and Bregs in healthy young and old C57Bl/6/Foxp3GFP mice. By the use of flow cytometry, the intensity of GFP fluorescence was analyzed in these mice, which was proportional to the level of Foxp3 transcription and allowed the identification of different Treg populations (Scirka, B. et al. Anti-GITR Antibody Treatment Increases TCR Repertoire Diversity of Regulatory but not Effector T Cells Engaged in the Immune Response Against B16 Melanoma. Arch. Immunol. Ther. Exp. (Warsz). 65, 553–564 (2017).
In this study we wanted to evaluate the effect of calcitriol in young and old mice, studying the same cell markers and cytokines which we found to be interesting in our previous studies on BALB/c mice bearing mammary gland cancer tumors. Our previous studies revealed differentiated impact of vitamin D analogs treatment in young (premenopausal) and old (postmenopausal) BALB/c mice. All cells populations, cytokines and cell markers studied here were selected based on our previous studies.
Round 2
Reviewer 1 Report
Comments and Suggestions for Authors
The authors much improved in the correct way the manuscript and now it could be suggested for publishing.